# COVID-19 at the Deep End: A Qualitative Interview Study of Primary Care Staff Working in the Most Deprived Areas of England during the COVID-19 Pandemic

**DOI:** 10.3390/ijerph18168689

**Published:** 2021-08-17

**Authors:** Claire Norman, Josephine M. Wildman, Sarah Sowden

**Affiliations:** Population Health Sciences Institute, Faculty of Medical Sciences, Newcastle University, Newcastle upon Tyne NE1 4LP, UK; josephine.wildman@newcastle.ac.uk (J.M.W.); sarah.sowden@newcastle.ac.uk (S.S.)

**Keywords:** COVID-19, health inequalities, general practice, primary care, social determinants of health, social prescribing, remote consulting, marginalised communities, health care inequalities, health/healthcare inequity

## Abstract

COVID-19 is disproportionately impacting people in low-income communities. Primary care staff in deprived areas have unique insights into the challenges posed by the pandemic. This study explores the impact of COVID-19 from the perspective of primary care practitioners in the most deprived region of England. Deep End general practices serve communities in the region’s most socioeconomically disadvantaged areas. This study used semi-structured interviews followed by thematic analysis. In total, 15 participants were interviewed (11 General Practitioners (GPs), 2 social prescribing link workers and 2 nurses) with Deep End careers ranging from 3 months to 31 years. Participants were recruited via purposive and snowball sampling. Interviews were conducted using video-conferencing software. Data were analysed using thematic content analysis through a social determinants of health lens. Our results are categorised into four themes: the immediate health risks of COVID-19 on patients and practices; factors likely to exacerbate existing deprivation; the role of social prescribing during COVID-19; wider implications for remote consulting. We add qualitative understanding to existing quantitative data, showing patients from low socioeconomic backgrounds have worse outcomes from COVID-19. Deep End practitioners have valuable insights into the impact of social distancing restrictions and remote consulting on patients’ health and wellbeing. Their experiences should guide future pandemic response measures and any move to “digital first” primary care to ensure that existing inequalities are not worsened.

## 1. Introduction

The COVID-19 pandemic has highlighted the impact of unfair and avoidable health inequalities, with death rates in deprived areas of the UK three times those of more affluent areas [1,2,3]. In this article, we make known the immediate and longer-term impacts of the pandemic from the perspective of primary care staff working in areas of blanket socioeconomic deprivation.

Governments around the world have used social distancing measures to slow the transmission of the virus. Social distancing aims to reduce virus transmission between households and includes physical distancing measures and the closure of sites of transmission, including schools and non-essential businesses. These measures are also referred to as “lockdowns”. Although lockdowns have been shown to be effective at suppressing the number of cases (and therefore deaths) from COVID-19 [4,5], concerns have been raised that the social distancing measures themselves are not without harm and that these harms will fall disproportionately on those living in disadvantaged circumstances [6]. These harms range from early consequences of people delaying medical assessments for COVID or non-COVID illness, to the longer-term effects of economic decline. In the UK, social distancing and lockdown policy has, at times, meant that schools have been closed to the majority of children and many services, including general practitioner (GP) surgeries, have attempted to reduce the amount of face-to-face contact they have with patients: appointments have become telephone or video by default, with 90% of consultations being via telemedicine [7]. The pandemic also necessitated the creation of “hot sites” for assessing patients with suspected COVID-19: these were often led by Primary Care Networks and staffed by local GPs [8]. Latterly, GPs have also co-ordinated community COVID-19 vaccine rollout.

North East England is the setting for this study. The North East of England has the lowest life expectancy in the country and in its most deprived areas, between the years 2010 and 2012 and the years 2016 and 2018, life expectancy has actually been decreasing [9]. The region performs the worst or second worst in the country for causes of death considered preventable: suicide and drug misuse at any age, as well as cancer, cardiovascular, respiratory, and liver diseases in those under 75. These differences can be attributed to a range of factors: high rates of poverty, poor housing, and the health and social sequelae of heavy industry and its more recent decline [10].

General Practitioners (GPs, also known as family practitioners or primary care practitioners), are the main providers of community care for acute and chronic illnesses in the UK. Everybody resident in or visiting the UK is entitled to access free medical care on the National Health Service (NHS) via their GP, although visitors may be charged for some hospital treatments or prescription medications. As one of a range of measures being implemented to address health inequalities, the North East and North Cumbria (NENC) is in the process of establishing a “Deep End” GP network of professionals working in practices in the region’s most deprived areas. Deep End practices serve populations living in areas of blanket deprivation with high proportions of patients living in the 15% most deprived local areas, based on postcode data. The NENC Deep End network was inspired by the GPs at the Deep End network in Scotland: a forum for advocacy, sharing ideas and developing interventions to mitigate health inequalities [11,12]. Since the founding of the original Scottish network in 2009, Deep End GP networks have been founded in several other regions of the UK, plus Ireland, Australia and Canada.

Social prescribing is a relatively new intervention which acknowledges the impact of the social determinants of health on people’s health and wellbeing. The aim is to use community organisations and other non-medical support services to address factors such as loneliness and poor housing, as an addition or alternative to offering clinical or pharmaceutical treatments to patients who may have multiple conditions or co-morbid mental and physical health problems [13]. The NHS England Long-Term Plan commits to providing social prescribing as part of its Universal Personalised Care model [14] and, typically, this is delivered by groups of GP practices via Primary Care Networks. The most common social prescribing model is for patients to be referred to a “link worker” (also known as care navigators or health trainers) who can identify the most appropriate service for their needs. Many of these link workers began working during the COVID-19 pandemic.

Deep End GPs and other primary care practitioners will have unique insights into the effect of COVID-19 on their communities, and the impact of public health measures designed to reduce viral transmission. In this qualitative study, conducted during the UK’s second wave of COVID-19, we aim to explore experiences of delivering primary care in a pandemic among staff working in practices in areas of blanket deprivation in North East England.

## 2. Materials and Methods

Data were collected as part of a wider project on the co-design of a Deep End network for the North East and North Cumbria. There was no direct patient or public involvement in this co-design project; however, the work was informed by a multi-agency steering group of policy and practice partners and researchers from across the NENC region.

Ethical approval was granted by Newcastle University research ethics committee (ref: 4322/2020).

### 2.1. Participants and Recruitment

Practices included in the core Deep End NENC network were identified using the methodology applied in the Scottish Deep End project. This entails identifying the proportion of each practice population living in the 15% most deprived areas of England, based on the Index of Multiple Deprivation (2019) and NHS Digital Practice Populations by Lower layer Super Output Areas (LSOA) (January 2020). All practices were ranked, and 34 North East and North Cumbria practices were found to be amongst the 10% most deprived practices in England against this measure.

A purposive framework was used to sample within the 34 NENC Deep End practices, prioritising geographical representation from Deep End practices across the NENC region. We also aimed to speak to participants with different levels of Deep End experience. Invitations to participate were sent via email to all Deep End practices, in addition to convenience and snowball sampling of participants known to the research team and purposive sampling of staff from practices that were the only Deep End practice in their locality. All staff members in Deep End practices were invited to attend.

Participants were sent a participant information leaflet and consent was gained by electronic completion of a form or by recording verbal consent at the start of the interview.

### 2.2. Data Collection

Interviews were carried out between October 2020–March 2021. In the temporal context of the COVID-19 pandemic, most interviews were undertaken in the early stages of the UK second wave. Cases were once again starting to rise, and lockdown restrictions (either local or national) were being tightened. Schools were open at this time and all but two interviews were prior to the approval of any COVID-19 vaccines.

Interviews were conducted by C.N. and J.M.W. and recorded using the Zoom video conferencing platform (Zoom Client for Meetings, Version 5.7.5 (939), Zoom Video Communications, Inc., San Jose, CA, USA). A topic guide was developed by the research team and used to provide a semi-structured approach, and this guide was continuously updated in line with emerging themes and participant feedback. Interview data were stored on a secure password protected server, accessible only by the research team.

### 2.3. Data Analysis

Interviews were auto-transcribed using Zoom video conferencing software, with manual corrections by C.N. C.N. and J.M.W. engaged in ongoing constant comparison of the data, allowing concurrent collection and analysis. Interviews were double coded to enhance validity. Thematic content analysis was used to code the transcripts and categorise them into emerging inductive themes [15]. NVivo (version 12, QSR International (UK) Ltd., Cheshire, UK) was used for data management and to support data coding [16].

## 3. Results

### 3.1. Participants

Fifteen interviews were carried out with primary care practitioners (Table 1): eleven GPs, two social prescribing link workers (LW), one nurse practitioner (NP) and one district nurse (DN). We spoke to participants with a breadth of experience: their careers in Deep End practices ranged between 3 months and over 30 years. All participants worked in urban areas, reflective of the areas of high blanket deprivation in the region. A wide geographic coverage was achieved; all but one of the North East’s Clinical Commissioning Group (CCG) areas were represented; at the time, CCGs were the organisations responsible for commissioning health services for individual geographical areas in England. A decision was taken to end recruitment as the emergence of recurrent themes suggested data saturation. We also found that the UK vaccine rollout was becoming a priority for primary care staff.

Our findings can be categorised into four overarching themes: (1) factors increasing the direct health risks of COVID-19 virus; (2) factors worsening pre-existing deprivation; (3) the role of social prescribing during COVID-19; (4) the benefits and costs of remote consulting.

### 3.2. Factors That Increased the Health Risks of COVID-19

Participants gave several reasons why COVID-19 cases and deaths might be higher in communities with high levels of deprivation. Multi-morbidity, rather than advanced age, was identified as the major risk factor for patients living in communities where *"getting to 55 would be pretty good"* (Interview 2, GP).

Concerns were raised about patients’ low levels of health literacy, which reduced their understanding of health messaging around COVID symptoms. As one GP noted, even widely publicised symptoms were not triggering patients to seek testing:


*“I still find it amazing, a guy I spoke to last week, cough and breathlessness: “do you think it could be COVID?” “Oh, I don’t know.” “Have you had a COVID test?” “No.” “Do you know how to get one?” “No, how do I do that doc?” And you just think, surely, with the last six months, the media, all the rest of it but it’d just not crossed his mind.”*
*Interview 8, GP*

Some participants noted that social distancing measures were not being adhered to in their communities. A lack of understanding, rather than deliberate rule flouting, was identified as a possible cause for non-observance:


*“I did a home visit yesterday, driving up the street, and we’re meant to not be socialising with anyone out of the household and I drove past about 13 people in a garden sharing a fag over the fence…You wonder how much of it is just like, I don’t want to follow it and how much is actually understanding the impact of your potential action.”*
*Interview 3, NP*

Patients’ ability to access to healthcare was identified as one of the most pressing challenges facing Deep End practices. One GP expressed concerns that the local “hot hub” facilities for assessing patients with suspected or confirmed COVID-19 in the community were inaccessible for those without cars.


*“The local hot site, say for patients in (local area) for COVID, you have to have a car to go…That does not help our patients”*
*Interview 13, GP*

Access issues created health risks for staff as well as patients. Lack of local testing and assessment services put pressure on GP surgeries to continue seeing symptomatic patients, potentially raising COVID case numbers among staff. Staff mentioned that their premises were smaller which made social distancing difficult.


*“The rooms are a lot smaller, patients are harder to manage on the phone because there’s lots of digital poverty, for example, people turn up and ask to be seen who don’t have a phone…So, I see a lot more people face-to-face here and probably as a result, I got COVID a month ago” *
*Interview 13, GP*

In addition to presenting a risk to staff health, exposure to the virus was creating staff shortages: 


*“We had a big outbreak of staff, having it in lockdown one, there was an entire team went off with it”*
*Interview 5, DN*


*“We had 11 members of staff off last week. Just complete carnage, trying to manage”*
*Interview 11, GP*

District nursing staff felt that they were being asked to take on extra responsibilities because some GPs were, understandably, trying to reduce patient contact.


*“It has been really hard to get GPs out to see anyone and a lot of the time we find as nurses that we’re telling GPs what we think over the phone, and they’re saying, “Okay, yeah, we’ll go with that” and not seeing the patient. So, even as far as palliative care—we’re having patients that haven’t been seen that are dying. And it’s been quite tough for the families because, you know, they would quite like to see a doctor…it does feel a bit like we are expected to diagnose someone so that they don’t have to visit.”*
*Interview 5, DN*

### 3.3. Factors Likely to Exacerbate the Effects of Deprivation in These Areas

Although recognising the need for lockdown measures, Deep End staff found that the social distancing measures were having a huge impact on their communities. Social distancing meant that community initiatives that practices had put in place were on hold, including social groups for isolated patients and group consultations for chronic pain.


*“Obviously, it’s all on hold at the minute because of COVID, which is making us all feel very uncomfortable because it became a bit of a lifeline really for some of our more isolated patients”*
*Interview 11, GP*

In addition to providing clinical care, supporting patients to address the social determinants of health formed a significant part of the workload in Deep End practices. The reduction in other services such as housing and social work was proving challenging for patients.


*“Housing is a recurring theme…that’s been really tricky recently, again because of COVID, it’s just (they) basically aren’t moving anybody. No matter what circumstances are, really they just will not move them.”*
*Interview 6, GP*

Concern around child safeguarding was a common theme, with participants reporting a reduction in family contact from health visitors and social workers. School closures also meant a reduction in safeguarding oversight, which was a source of deep concern to practice staff.


*“The child safeguarding situation fills me with dread…throughout lockdown because we’ve been one of the services that’s remained open and visible, we’re being presented with a lot of this stuff which is difficult. And we’re being presented with it without lots of the support that we normally have to manage it.”*
*Interview 1, GP*

Although telephone contact was happening, this was not felt to be adequate.


*“Health visitors are not doing a lot of face-to-face visits … there’s always the risk that we’re missing things because they’re not being seen face-to-face—it’s just been telephone.”*
*Interview 2, GP*

The long-term effects of the pandemic were also a significant concern: communities that were already struggling economically may not be able to recover.


*“The legacy of this, the unemployment, the deprivation, that’s just going to get worse for patients because as with all of these things, our communities will be the hardest hit going forward. They’re not going to recover. They’re not going to bounce back…in the way that other areas may be able to. And so, yeah, it kinda depresses me really ‘cos I just think this is just going to make things worse…That’s my concern from COVID, is just it’s just going to push these communities further down.”*
*Interview 11, GP*

### 3.4. Social Prescribing during COVID-19

A lot of social prescribing services in the North East of England were commissioned just before or during the COVID-19 pandemic. Their roles were constantly changing.


*“For the most part we started in the pandemic. So, a lot of it was initially just COVID response stuff. Yeah, you know. Food parcels, medication deliveries, that kind of thing and like check in calls really for vulnerable patients. So initially we were getting sent a list of COVID patients or 80 plus (year old patients) for example and pretty much cold calling them. You know, we phoned on behalf of the GP practice. “Is there anything you need in this lockdown?” that kind of thing. But we’ve moved away from that.”*
*Interview 14, Link worker*

Lockdown periods saw an increase in referrals, particularly for mental health difficulties.


*“There’s a huge amount people getting referred for support with losses and also just anxiety, generally coping with the lockdown, problems with isolation, no contact with families and I think people felt were moving beyond that in the autumn of last year and then to go into another lockdown. I think a lot of people really dipped during the winter with their mental health.”*
*Interview 15, LW*

Increased demand plus a reduction in other services due to the pandemic meant that waiting lists were often long, particularly for talking therapies. The social prescriber was seen as a stop gap for patients who were needing extra support *“We’re doing a lot of more long-term handholding at the moment”* said one link worker (Interview 14, LW).

However, *“bearing in mind, we’re not trained counsellors or therapists”* (Interview 14, LW), social prescribing link workers in the Deep End were proving a valuable resource for patients with non-clinical needs during the pandemic. One GP observed:


*“They’ve done a lot of very intense work with quite complex and risky people. So they’ve contained a lot of that complexity and risk, which I hadn’t really appreciate that they’d be doing.”*
*Interview 1, GP*

However, the link workers themselves found that the provision of adapted services to refer patients to was very variable.


*“(North East (NE) town) seems to have a lot more going on if you like, a lot more of their groups seem to have adapted to the pandemic, so they’re offering remote sessions or virtual sessions instead. Whereas in (another NE town) a lot of things seem to have just ceased”*
*Interview 14, LW*

They also found that many referring practitioners and patients had great expectations of what the service could offer, which often did not reflect reality.


*“You know there’s a lot of sort of magic wand expectations, and you know the GP saying “right, you’re really lonely or isolated. You stuck at home. I’m gonna refer you social prescribing because they can help.”…We got to be able to do this, but those aren’t options at the moment, so we need to look at something else. And I think a lot of the time that could be quite disappointing. Because they’re like, “well, you know my GP said that you could help me and, you know, get me out and about and things like that.””*
*Interview 14, LW*

### 3.5. Benefits and Costs of Remote Provision

Modifications to the way of working brought some positives and some participants felt that the pandemic acted as a catalyst for change.


*“COVID’s been a shot in the arm to make changes that we, you know, have been considering for a while anyway, like changes to our access system”*
*Interview 1, GP*

One participant felt strongly that the increased use of technology was a positive and that the pandemic had just brought forward an inevitable move away from face-to-face consulting.


*“Everybody’s doing telephone consultation. You don’t need to push anymore. Video. Yeah, everybody’s trying to do video consultation, and certainly once they know how to do it. So, I don’t want to go back to the old days.”*
*Interview 10, GP*

Text messaging was felt to be particularly useful, especially for reaching patients who were sometimes difficult to contact.

The social prescribing link workers found remote working meant they could support more people, because they were not spending so much time travelling; however, this came at the cost of fully assessing their circumstances.


*“I mean, you are not travelling, so you’ve got more time....So just concentrate on having that contact with people because you’re literally picking up the phone and you know you’re not having to drive to them, but I think it’s really hard to get a picture of people’s situations over the phone.”*
*Interview 15, LW*

While there was enthusiasm for the potential benefits of increased use of technology, this was tempered by an understanding of the risks of digital exclusion. Although participants understood that reducing physical contact was important during the pandemic, there was concern that the move to “remote by default” consulting may persist. Participants were keen to make sure that any change to consulting methods was not disadvantaging those who already struggled to access healthcare.


*“If nothing else, making sure that what, anything we introduce doesn’t disadvantage those already disadvantaged.”*
*Interview 8, GP*

The drive for increased remote provision was felt to be a political decision, made without adequate knowledge of the challenges that deprived communities faced.


*“In terms of the technology that Matt Hancock seems to think is the way forward and just because him and all his peers, you know, have access to all the technology and it’s very convenient for them to consult with their GP via zoom, that is not how it is for the people where I work.”*
*Interview 11, GP*

Online solutions were not felt to be accessible for many patients in the Deep End.


*“It’s so hard with COVID, because I could say if there was more…befriending schemes and things, but at the moment, it’s all kind of zoom based and none of our patients could really do that…I don’t think many of them have smartphones or laptops.”*
*Interview 5, DN*

Digital poverty and lack of Information Technology (IT) literacy were often mentioned as concerns in the move to remote consulting. Participants reported that older patients in the communities often did not have internet-enabled devices at all and younger patients may have had devices, but Wi-Fi or data access was variable. Video consulting technology was available but rarely used.


*“I think IT literacy, access to technology, equipment, Wi-Fi, that’s been a challenge for us because video consults are often not an option. Even sometimes people who don’t even have phones or are unable to take photos, that’s made it difficult in terms of COVID and remote consulting.”*
*Interview 7, GP*

While some initiatives have attempted to move online, this was felt to be problematic.


*“If you do things via Zoom you’re then immediately removing a group of the population who can’t be involved.”*
*Interview 8, GP*

Social prescribing link workers found that, although some services had moved online, there was mixed enthusiasm for this.


*“We’ve gone to a digital offer but then that’s only accessible to so many people, so as much as we can encourage people to become like digitally active like, not everyone wants to. People want that face-to-face contact.”*
*Interview 15, Link worker*

Remote solutions were also problematic for patients with language barriers:


*“I find it really hard with people via an interpreter, trying sometimes to assess what’s going on...as opposed to when you just see them face-to-face, it’s a lot easier.”*
*Interview 13, GP*

Potentially most seriously, some felt the reduction in face-to-face contact was going to irreparably damage the relationship between primary care and the community.


*“I think barriers are going up because people don’t have the technology. I think barriers are going up because people like to see their GP you know, and I suspect that’s even more so in deprived areas…I think for some people we are a bit of the centre of that community and I think, you know, you put barriers up in that we’re saying to them all “well you probably shouldn’t be going to the surgery, we’ll try and do this over the phone or I’ll send something...” you know it’s not good.”*
*Interview 11, GP*

## 4. Discussion

This paper adds important context to the quantitative data on excess morbidity and mortality in deprived populations during the COVID-19 pandemic. We identify mechanisms through which socioeconomic deprivation exposes both patients and healthcare providers to an increased risk of COVID-19. We also add to the literature on the harms associated with some public health measures by highlighting the role of primary care in addressing the social determinants of health and the ways in which the pandemic is likely to worsen existing deprivation. We also highlight the work done by district nursing and social prescribing link workers during the pandemic. Finally, we contribute to the conversation around the move to remote consulting in primary care by identifying the potential risks that the drive to digital-first care poses to the provision of primary care in areas of deprivation.

Reasons for higher morbidity and mortality rates in deprived communities are multifactorial. People in deprived communities are more likely to be exposed to the virus, through overcrowded housing in built-up urban areas and through work in low-paid key worker roles [17]. As noted by the practitioners in our research, pre-existing vulnerability to the disease may be higher, due to increased rates of underlying health conditions such as smoking-related respiratory disease and hypertension [1]. The QCovid risk prediction model considers deprivation alongside other risk factors, such as ethnicity, body mass index and underlying health problems [18]. We welcome the UK government’s decision to use the model to identify high-risk patients for shielding and vaccination: not only should this reduce COVID-19 morbidity and mortality in deprived areas, but it has set a precedent for publicly acknowledging the negative health impacts of deprivation. Low health literacy and ever-changing guidelines may lead to underestimation of the risks of the disease [19] which was also reflected in our findings: improving health literacy among the general population should be an urgent health and education priority.

We add that the location of COVID-19 assessment centres may make them inaccessible to low-income households without cars: this should also be considered for test centres and vaccination hubs. Without access to these sites, patients are more likely to visit their usual GP surgery and potentially increase transmission among staff and other patients. Risks to staff health are particularly important given that GP surgeries in deprived areas are likely to have fewer GPs per population, and that those who work there are more likely to be older [20] and, therefore, more vulnerable to becoming seriously unwell with COVID-19. Our participants also raise concerns that smaller premises made social distancing more challenging, potentially contributing to rates of COVID-19 transmission within the surgery. We highlight the need for healthcare spaces to have the space and ventilation required to minimise spread of diseases similar to COVID-19.

We add to the literature on the negative consequences of social distancing and remote working, highlighting the challenges that GPs faced as one of the few visible public services that remained functioning near-normally throughout the pandemic. Our findings highlight the vital role general practice plays in supporting patients with their social needs, but also the need for specific enhanced support in this area. Already vulnerable and socially isolated patients became increasingly so, with the cessation of many support services or organisations, or a move to online technologies that were not accessible for Deep End patients. However, social prescribing link workers were praised for their work, telephoning vulnerable people and navigating their non-medical needs. Concerns around child safeguarding were not unfounded, with a significant reduction in the number of children referred for Child Protection Medical Examinations during the period of the first lockdown [21,22]. A recent report highlighting Child Welfare Inequality shows the unequal distribution of safeguarding interventions across the socioeconomic spectrum and the need for enhanced safeguarding support in deprived communities during and after the pandemic [23]. As we increasingly move to remote and digital services, the value of home visiting and face-to-face encounters for picking up on signs of struggling families, or neglect and abuse, must not be forgotten.

Social prescribing is often seen as a panacea for many of the NHS’s problems: supporting complex patients while increasing the amount of GP and nurse time for more traditional medical problems. Our link worker participants describe the challenges of starting new roles during a pandemic and the uncertainty and limited availability of onward and ancillary services, plus the expectation from professionals and patients that they were going to be able to wave a “magic wand”. Their experiences of trying to carve out the role is similar to that described by Frostick and Bertotti [24], but likely made even more challenging by the nature of working with complex patients in the Deep End setting [25]. Primary care staff and commissioners should be clear with what they expect social prescribing to achieve in their local area and provide appropriate levels of oversight and supervision.

Our participants’ experience of remote consulting matched some of those of Flemish GPs in the early part of the pandemic: fears around missed diagnoses and providing suboptimal care [26]. We add a Deep End perspective to the comprehensive review of Murphy et al., of the move to remote consulting in UK general practice [7]. Digital poverty and lack of IT literacy are major concerns for primary care staff in deprived communities and should be considered as a priority when taking forward the NHS long-term plan for a digital-first primary care [27]. Improving IT literacy among older or vulnerable populations should be considered a necessity to make sure that no-one is excluded as healthcare moves into the digital sphere; collaboration between adult education and health services may be required. Access to internet-enabled devices and affordable mobile data or Wi-Fi will also be vital.

Practitioners in these areas are aware that their surgery can often be a centre of the community. Whether this is an appropriate role for general practice, or possibly a symptom of the general decline of social capital, our participants worry that a move to remote consulting will damage this relationship. These concerns are reflected in research into the portrayal of remote consulting in the media during the pandemic, which showed a decline in popularity and acceptance between the first and second waves of COVID-19 in 2020 [28]. A recent systematic review found that telephone consulting was favoured by certain groups: women, younger people, very old people and non-immigrants. Similarly, online consulting was weakly associated with younger, more affluent and educated populations [29]. It is vital that the concerns of practitioners in areas of deprivation are reflected in changes to consulting so that the nature of community medicine is not irreversibly altered.

Our district nurse participant describes her discomfort at being asked to act beyond her usual role. This contrasted with the experiences of our GP participants who felt that the reduction in the number of home visits was patient-led. Although inter-practice variability is likely, we suspect that the district nursing views had not been directly sought. Little has been written in the peer-reviewed literature about the experiences of community nurses during the pandemic, but the Royal College of Nurses surveyed its District and Community Nursing members and found similar themes to those raised by our participant [30]. Bowers et al. raise similar concerns, particularly around palliative care, adding that the work of community teams has often been overlooked amid widespread media coverage of those working in hospital [31]. Although a reduction in GP home visits was felt to be necessary to reduce virus transmission, it is important that relationships are not damaged between doctors and their district nursing colleagues, or patients and their relatives. There may also be implications for patient care and diagnosis if nurses and other allied health professionals are not adequately supported. Macdonald et al. argue that home visits for vulnerable and end-of-life patients must remain a priority for GPs as they are the experts in continuity and overseeing complexity [32]. These experiences are unlikely to be specific to deprived communities, but they highlight the need for multidisciplinary pandemic response planning.

### 4.1. Strengths and Limitations

Most interviews were carried out by C.N., a GP registrar who worked in a Deep End practice during the first wave. This had a positive impact on accessing participants and was felt that this resulted in interviews that were more candid, as demonstrated in the richness of the data. There was also increased understanding of clinical terminology and local systems. A topic guide was used to avoid shared conceptual blindness and reduce the risk of biasing the agenda with C.N.’s personal experience or opinions [33].

Conducting interviews via video proved acceptable and convenient, as well as COVID-safe [34]. Participant recruitment was likely adversely affected due to the increasing clinical burden on primary care staff during the second wave of COVID-19 in late 2020 and early 2021, particularly during the vaccine rollout.

### 4.2. Implications for Practice and Research

Our findings are relevant to policymakers in both Primary Care and Public Health: it is vital that any public health intervention is ethically implemented, with consideration given to the most vulnerable in society [35]. Care should be taken not to increase existing inequality. They are also relevant to local authorities and adult education teams who contribute to the wider social determinants of health and the growing need for IT literacy.

Although we are pleased to have included the views of nursing staff and social prescribing link workers, future research should include other primary care practitioners, such a health visitors and midwives. Qualitative research with people who live in areas with high rates of COVID-19 infections and deaths would provide even more insight into the potential reasons behind the variation along socioeconomic lines. Further data should be collected on the benefits of social prescribing for patients and other NHS staff during the COVID-19 pandemic.

## 5. Conclusions

Deprived communities are facing the brunt of the COVID-19 pandemic. Through the eyes of primary care staff in these communities, we have shown that this goes beyond the impact of the disease itself, with social distancing measures and remote consulting exacerbating many existing inequalities. Deep End primary care practitioners are well-placed to advocate for their patients and their views are crucial in ensuring that future Public Health measures and major systems changes are implemented in ways that reduce rather than maintain or even increase existing inequalities in health and healthcare.

## Figures and Tables

**Table 1 ijerph-18-08689-t001:** Participant characteristics.

Characteristic		*N*
Gender		
	Male	7
	Female	8
Occupation		
	General Practitioner (GP) partner	8
	Salaried GP	3
	Social prescribing link worker (LW)	2
	Nurse practitioner (NP)	1
	District nurse (DN)	1
Time spent working in the Deep End		
	0–3 years	5
	4–9 years	3
	10–20 years	3
	21–31 years	4

## Data Availability

The data presented in this study are available on request from the corresponding author. The data are not publicly available due to containing potentially identifiable information about participants’ places of work.

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
