# Peer review of "COVID-19 at the Deep End: A Qualitative Interview Study of Primary Care Staff Working in the Most Deprived Areas of England during the COVID-19 Pandemic"

_ijerph, 2021, doi:10.3390/ijerph18168689_

Round 1

Reviewer 1 Report

Thank you for inviting me to review this interesting and well written paper. It highlights many important issues for policy and practice, both for the future and during the current pandemic.

I have a few minor comments which I hope will help the authors strengthen the methods section further. They state that they used a purposive framework for sampling, which is entirely appropriate. However, the only characteristic they mention is geographical representation. Were other characteristics taken into account (e.g. mix of GPs and other healthcare practitioners or length of career in Deep End practices)? Given the importance of the nursing workforce in primary care, the inclusion of more participant in nursing roles would have strengthened the study but I can fully understand why the authors had to cease data collection when the vaccination roll-out began. Perhaps it would be helpful to reflect on the implications of this in the strengths and weaknesses section.

I felt the approach to the data analysis was very strong, in particular the double coding, and the use of thematic content analysis was entirely appropriate. I just had one question (which I do not necessarily expect the authors to respond to) – was the analysis entirely inductive or did the authors have some broad pre-existing themes which they wanted to explore?

I hope these comments are helpful. 

Author Response

Thank you for your comments and I'm pleased you enjoyed the paper!

Point 1:

Were other characteristics taken into account (e.g. mix of GPs and other healthcare practitioners or length of career in Deep End practices). Given the importance of the nursing workforce in primary care, the inclusion of more participant in nursing roles would have strengthened the study but I can fully understand why the authors had to cease data collection when the vaccination roll-out began.

Response 1:

Invitations to take part in the research made it clear that all members of the primary care team would be valued participants. I have added this to 2.1 Participants and recruitment. We managed to speak to people with a wide range of career lengths and I have emphasised this in 3.1 Participants.

Point 2:

Was the analysis entirely inductive or did the authors have some broad pre-existing themes which they wanted to explore?

Response 2:

The analysis of the COVID-related section of the interviews was entirely inductive. I have added the word inductive to this section

Many thanks.

Reviewer 2 Report

Interesting qualitative study about the experiences of 'Deep End' primary care practitioners during the COVID-19 pandemic .  Very well written.  Particularly strong discussion section and I was pleased to see the important insights from the district nurse participant set within a wider context.  No major comments.

Some typos noted (e.g. p5, line 200: "a bit the lifeline" and line 215: "we're being be presented with" among others.)

Author Response

Thank you for your kind comments and I am pleased you enjoyed the paper.

Point 1:

Some typos noted (e.g. p5, line 200: "a bit the lifeline" and line 215: "we're being be presented with" among others.)

Response 1:

These have been corrected.

Reviewer 3 Report

Thank you for the opportunity to read this excellent paper. I attach some suggestions for its improvement.

Author Response

Thank you very much for your considered comments. You are clearly an advocate for the Deep End! 

Point 1:

Occupational risk: Did the authors collect any information from their respondents on what actions they thought were necessary to reduce risk of exposure to COVID-19? For example, improved ventilation in the practice, more space, etc? If so, could this be included in the report? I think there is a general need to improve ventilation of living and working spaces in the UK – I have lived and worked in London and found it to be a grim and stuffy environment compared to other countries I have lived – and I think it is important to report on these issues where possible, so if the information is available it would be helpful to report. 

Response 1:

We did not specifically ask about ways to reduce risk. On page 5, line 180-185 a participant mentions the fact that his Deep End practice rooms are smaller and that may have contributed to him contracting COVID. You make an excellent point and I have added this to the discussion. 

Point 2:

Adult education: As never before this pandemic has shown that the need to improve digital literacy in the elderly is no longer a boutique request, but an essential element of their inclusion in society, and widespread improvement in digital literacy in older people is going to require major changes in adult education policy. It would be good to highlight the consequences of this illiteracy and issue a call to improve it

Response 2:

I have added to the discussion regarding this, and also on health literacy in general.  

Point 3:

Infrastructure: Boris Johnson is fond of talking about “leveling up” the North and it is in the news now, but the authors highlight woeful shortcomings in housing, access to basic IT services (including phones and internet connectivity), public transport and health workforce which need to be addressed. I would like to see them discuss the fatal consequences of poor infrastructure in these Deep End communities and make recommendations about what needs to be done to improve pandemic (and seasonal influenza) resilience in poor areas.

Response 3:

I have added a section about the need for access to IT devices and wifi, however our participants did not specifically mention housing and this is covered in more detail in other referenced articles (Corris et al 2020, Bambra et al 2020). 

Point 4:

Child safeguarding: If future healthcare is going to be more digital, what specific recommendations can the authors glean from their interviews about how to provide effective safeguarding services?

Response 4:

I have added a sentence on the value of face to face contact and home visiting with regard to assessing circumstances and safeguarding risk. 

Point 5:

The role of the practice: It doesn’t seem right to me that the GP practice should be the centre of the community during a pandemic, as described in this paper, and I think the implications of this for the future of Deep End communities needs to be explored. Does this reflect poor social capital and/or the collapse of other more appropriate avenues of social solidarity in these areas? Is the NHS the only surviving functional public service in these areas? If so, this is surely not right, and here the authors have an opportunity to call for the rejuvenation of social and public life in these communities.

Response 5:

We cover this in more detail in our forthcoming paper ‘co-designing a Deep End network for the North East and North Cumbria.’ I have made a brief comment on whether or not the GP surgery being at the centre of the community is appropriate. 

Point 6:

I feel like I say this a lot in peer reviews, but I would like to see the discussion strengthened with specific calls to action and recommendations, as well as criticism of existing structures. These communities are Deep End communities for a reason!

Response 6:

I hope that I have made stronger calls to action as detailed above. 

Point 7:

I think in the introduction, for non-UK readers, a little more detail or information about the health care system (the role of GPs, what is a clinical commissioning group, etc) would be helpful.

Response 7:

I have added some extra information about GPs and CCGs to the introduction and ‘participants’ section of the findings. 

Point 8:

In line 62 the authors refer to “the” NENC, but I think there is a word missing here? I think it should be “the NENC commissioning group” or “the NENC health service” or something?

Response 8:

NENC stands for “North East and North Cumbria” – however, the region that is the North East of England is often referred to as “the North East” so this is how it should read.